

# Prospects of measurements on the anomalous magnetic and electric dipole moments of the $\tau$-neutrino in $pp$ collisions at the LHC

Alejandro Gutiérrez-Rodríguez[1][*], Murat Köksal[2],
Ahmet A. Billur[3] and María de los Ángeles Hernández-Ruíz[4]

**1** Unidad Académica de Física, Universidad Autónoma de Zacatecas,
Apartado Postal C-580, 98060 Zacatecas, México.
**2** Deparment of Optical Engineering, Cumhuriyet University, 58140, Sivas, Turkey.
**3** Deparment of Physics, Cumhuriyet University, 58140, Sivas, Turkey.
**4** Unidad Académica de Ciencias Químicas, Universidad Autónoma de Zacatecas,
Apartado Postal C-585, 98060 Zacatecas, México.

[*] alexgu@fisica.uaz.edu.mx

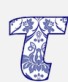

*Proceedings for the 15th International Workshop on Tau Lepton Physics,
Amsterdam, The Netherlands, 24-28 September 2018*

## Abstract

In this paper the production cross-section $pp \rightarrow (\gamma, Z) \rightarrow \nu_\tau \bar{\nu}_\tau \gamma + X$ in $pp$ collisions at $\sqrt{s} = 13, 14, 33 \ TeV$ is presented. Furthermore, we estimate bounds at the 95% C.L. on the dipole moments of the $\tau$-neutrino using integrated luminosity of $\mathcal{L} = 100, 500, 1000, 3000 \ f \ b^{-1}$ collected with the ATLAS detector at the LHC and we consider systematic uncertainties of $\delta_{sys} = 0\%, 5\%, 10\%$. We found that the current and future LHC bounds are weaker than other experiments reported in previous investigations. However, it is shown that the estimated process is a good prospect for probing the dipole moments of the $\tau$-neutrino at the LHC.

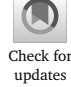
## 1 Introduction

In the Standard Model (SM) the neutrinos are massless particles whose electromagnetic property are poorly known experimentally. In addition, the observation of neutrino oscillation shows the necessity of neutrino masses, which implies that the SM to be modified such that non-trivial electromagnetic structure of neutrino should be reconsidered [1]. In the minimal extension of the SM to incorporate the neutrino mass the anomalous magnetic moment (MM) of the neutrino is known to be developed in one loop calculation, $\mu_\nu = \frac{3eG_F m_{\nu_i}}{(8\sqrt{2}\pi^2)} \simeq 3.1 \times 10^{-19}(\frac{m_{\nu_i}}{1 \ eV})\mu_B$, where $\mu_B = \frac{e}{2m_e}$ is the Bohr magneton [2], and the non-zero mass of the neutrino is essential to get a non-vanishing magnetic moment. Furthermore,

the SM predicts CP violation, which is necessary for the existence of the electric dipole moments (EDM) of a variety physical systems. The EDM provides a direct experimental probe of CP violation [3], a feature of the SM and beyond SM physics. The signs of new physics can be analyzed by investigating the electromagnetic dipole moments of the tau neutrino, such as its MM and EDM.

The physics program of the ATLAS Collaboration at the LHC [4] contemplates the study of the hadroproduction of $Z$ bosons associated with one or two photons. To carry out this study the ATLAS Collaboration use $\mathcal{L} = 20.3\, fb^{-1}$ of $pp$ collisions collected with the detector operating at a center-of-mass energy of $\sqrt{s} = 8\, TeV$. For their analysis they use the decays $Z/\gamma^* \to l^+l^-$ with $l = (e^-$ or $\mu)$ and $Z \to \nu\bar{\nu}$. The production channels studies are $pp \to l^+l^-\gamma + X$ and $pp \to l^+l^-\gamma\gamma + X$. Other important channels are $pp \to \nu\bar{\nu}\gamma + X$ and $pp \to \nu\bar{\nu}\gamma\gamma + X$.

Motivated for the physical program of the ATLAS collaboration with regard to the study on the dipole moments of the neutrino, in this paper we explore the possibility of probing the dipole moments of the tau-neutrino through the process $pp \to (\gamma, Z) \to \nu_\tau \bar{\nu}_\tau \gamma + X$ for $\sqrt{s} = 8, 13, 14, 33\, TeV$. We use integrated luminosity of $\mathcal{L} = 20, 50, 100, 200, 500, 1000, 3000\, fb^{-1}$ collected with the ATLAS detector at the LHC and we consider systematic uncertainties of $\delta_{sys} = 0\%, 5\%, 10\%$. All our study was carried out with a 95% confidence level (C.L.). It is shown that the theorized process is a good prospect for probing the dipole moments of the tau-neutrino at the LHC. Although our study was carried out only with data from the ATLAS Collaboration [4], it is worth mentioning that similar results are obtained when considering the data of the CMS Collaboration [5].

## 2 Production process $\nu_\tau\bar{\nu}_\tau\gamma$ in $pp$ collisions

Theoretically the electromagnetic properties of neutrinos best studied and well understood are the MM and the EDM. Despite that the neutrino is a neutral particle, neutrinos can interact with a photon through loop (radiative) diagrams. However, a convenient way of studying its electromagnetic properties on a model-independent way is through the effective neutrino-photon interaction vertex which is described by four independent form factors. The most general expression for the vertex of interaction $\nu_\tau\bar{\nu}_\tau\gamma$ is given by [6,7]

$$\Gamma^\alpha = eF_1(q^2)\gamma^\alpha + \frac{ie}{2m_{\nu_\tau}}F_2(q^2)\sigma^{\alpha\mu}q_\mu + \frac{e}{2m_{\nu_\tau}}F_3(q^2)\gamma_5\sigma^{\alpha\mu}q_\mu + eF_4(q^2)\gamma_5\left(\gamma^\alpha - \frac{\not{q}q^\alpha}{q^2}\right), \quad (1)$$

where $e$ is the charge of the electron, $m_{\nu_\tau}$ is the mass of the tau-neutrino, $q^\mu$ is the photon momentum, and $F_{1,2,3,4}(q^2)$ are the electromagnetic form factors of the neutrino. In general the $F_{1,2,3,4}(q^2)$ form factors they are not physical quantities, but in the limit $q^2 \to 0$ they are quantifiable and related to the static quantities corresponding to charge radius, magnetic moment (MM), electric dipole moment (EDM) and anapole moment (AM), respectively [7]. In this paper we are interested in the anomalous magnetic moment $\mu_{\nu_\tau}$ and the electric dipole moment $d_{\nu_\tau}$, which are defined in terms of the $F_2(q^2 = 0)$ and $F_3(q^2 = 0)$ form factor as follows:

$$\mu_{\nu_\tau} = \left(\frac{m_e}{m_{\nu_\tau}}\right)F_2(0)\mu_B, \quad (2)$$

$$d_{\nu_\tau} = \left(\frac{e}{2m_{\nu_\tau}}\right)F_3(0). \quad (3)$$

### 2.1  $pp \rightarrow \nu_\tau \bar{\nu}_\tau \gamma + X$ cross-section beyond the SM

We study the process of simple production of photon in association with a pairs of massive neutrinos that could be observed at the LHC, the schematic diagram is given in Fig. 1. The double production of $\nu_\tau$ may take place due to the reaction

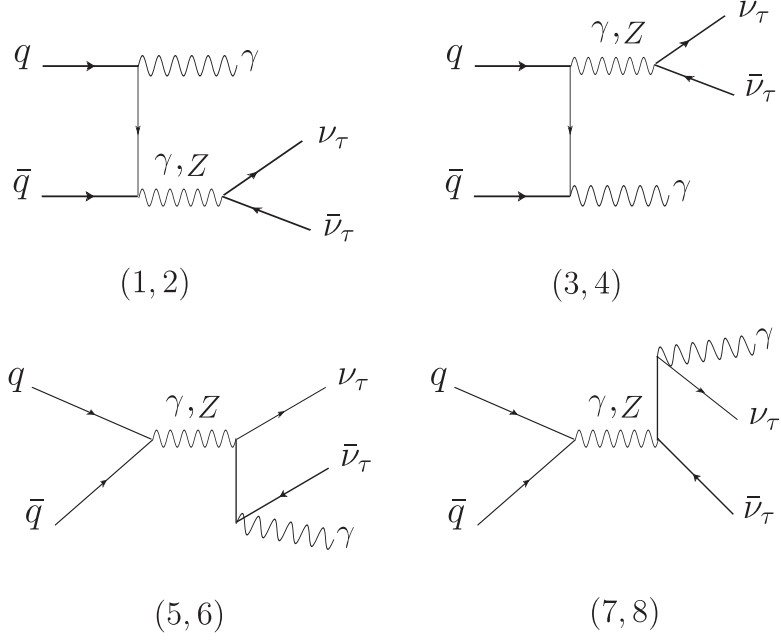

Figure 1: The Feynman diagrams for the processes $q\bar{q} \rightarrow (\gamma, Z) \rightarrow \nu_\tau \bar{\nu}_\tau \gamma$.

The Feynman diagrams Fig. 1 of the process

$$q\bar{q} \rightarrow (\gamma, Z) \rightarrow \nu_\tau \bar{\nu}_\tau \gamma. \tag{4}$$

We present numerical fit functions for the total cross-sections in relation to center-of-mass energy and in terms of the form factors $F_2$ and $F_3$.

- For $\sqrt{s} = 14\, TeV$.

$$
\begin{aligned}
\sigma(F_2) &= \left[ (3.463 \times 10^{12}) F_2^4 + (24582) F_2^2 + 0.022 \right] (pb), \\
\sigma(F_3) &= \left[ (3.463 \times 10^{12}) F_3^4 + (24582) F_3^2 + 0.022 \right] (pb).
\end{aligned}
\tag{5}
$$

- For $\sqrt{s} = 33\, TeV$.

$$
\begin{aligned}
\sigma(F_2) &= \left[ (2.645 \times 10^{13}) F_2^4 + (88985) F_2^2 + 0.0627 \right] (pb), \\
\sigma(F_3) &= \left[ (2.645 \times 10^{13}) F_3^4 + (88985) F_3^2 + 0.0627 \right] (pb).
\end{aligned}
\tag{6}
$$

In the expressions for the total cross-section Eqs. (5)-(7), the coefficients of $F_2(F_3)$ given the anomalous contribution, while the independent terms of $F_2(F_3)$ correspond to the cross-section at $F_2 = F_3 = 0$ and represents the SM cross-section magnitude.

### 2.2  Bounds on the anomalous couplings at the LHC

We apply the following cuts to reduce the background and to optimize the signal sensitivity:

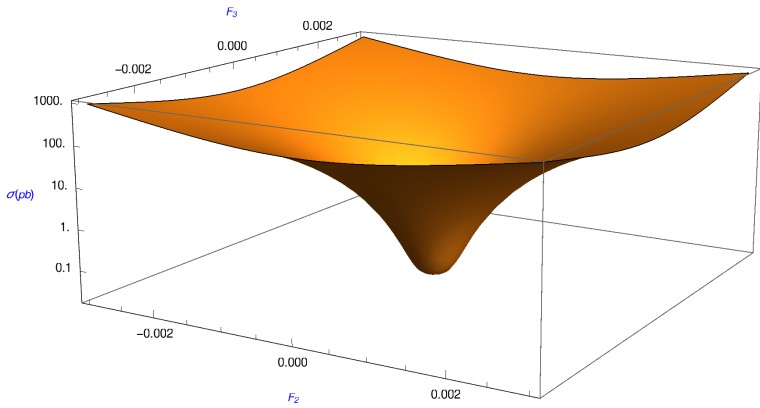

Figure 2: The total cross-sections of the process $pp \to \nu_\tau \bar{\nu}_\tau \gamma + X$ as a function of $F_2$ and $F_3$ for center-of-mass energy of $\sqrt{s} = 13\,TeV$.

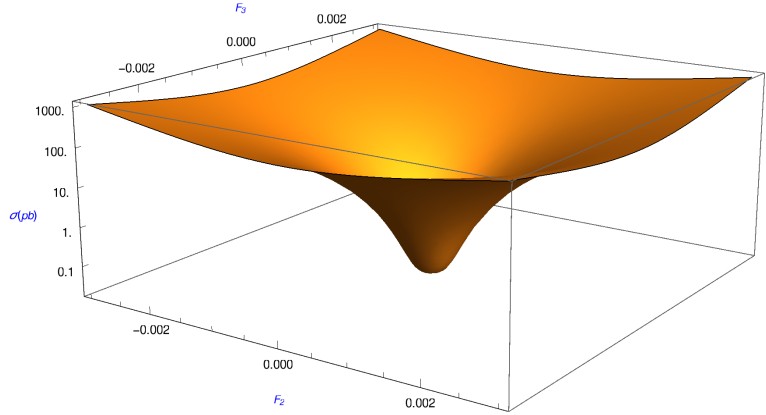

Figure 3: Same as in Fig. 2, but for $\sqrt{s} = 14\,TeV$.

$$
\begin{aligned}
E_T^\gamma &> 150\,GeV, \\
p_T^{(\nu,\bar{\nu})} &> 150\,GeV, \\
|\eta^\gamma| &< 2.37,
\end{aligned}
$$

The best bounds obtained on $\mu_{\nu_\tau}$ and $d_{\nu_\tau}$ are $1.251 \times 10^{-6}\,\mu_B$ and $2.424 \times 10^{-17}\,ecm$, respectively, as shown in Table I. These limits are the most stringent to date, which are obtained through process $pp \to \nu_\tau \bar{\nu}_\tau \gamma + X$ and with the parameters of the LHC.

## 3 Conclusions

In conclusion, we have study the possible manifestation of the MM and the EDM of the tau-neutrino in collisions $pp$ using the ATLAS detector at the LHC. The channel $pp \to (\gamma, Z) \to \nu_\tau \bar{\nu}_\tau \gamma + X$ yields the most stringent limits on the $\mu_{\nu_\tau}$ and $d_{\nu_\tau}$ set at a hadron collider to date: $\mu_{\nu_\tau} = 1.251 \times 10^{-6}\,\mu_B$ and $d_{\nu_\tau} = 2.424 \times 10^{-17}\,ecm$. This is roughly a factor of 2.63 improvement over the result published for the L3 Collaboration in Ref. [8]. In addition, other limits on the $\nu_\tau$ MM and EDM in different context are reported in Refs. [9–19].

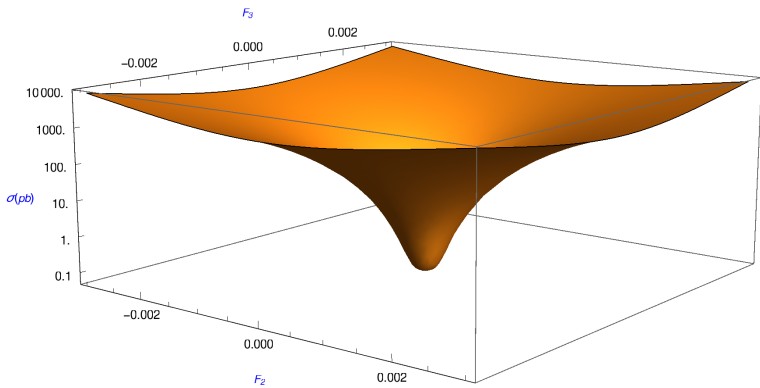

Figure 4: Same as in Fig. 2, but for $\sqrt{s} = 33\,TeV$.

Table 1: Sensitivity on the $\mu_{\nu_\tau}$ magnetic moment and the $d_{\nu_\tau}$ electric dipole moment for $\sqrt{s} = 33\,TeV$ and $\mathcal{L} = 100, 500, 1000, 3000\,fb^{-1}$ at 95% C.L. through the process $pp \to \nu_\tau \bar{\nu}_\tau \gamma + X$.

| | $\sqrt{s} = 33\,TeV,$ | 95% C.L. | |
|---|---|---|---|
| $\mathcal{L}(fb^{-1})$ | $\delta_{sys}$ | $\lvert\mu_{\nu_\tau}(\mu_B) \times 10^{-6}\rvert$ | $\lvert d_{\nu_\tau}(ecm)\rvert$ |
| 100 | 0% | 2.196 | $4.256 \times 10^{-17}$ |
| 100 | 5% | 3.298 | $6.391 \times 10^{-17}$ |
| 100 | 10% | 3.960 | $7.674 \times 10^{-17}$ |
| 500 | 0% | 1.704 | $3.303 \times 10^{-17}$ |
| 500 | 5% | 3.275 | $6.347 \times 10^{-17}$ |
| 500 | 10% | 3.953 | $7.661 \times 10^{-17}$ |
| 1000 | 0% | 1.518 | $2.942 \times 10^{-17}$ |
| 1000 | 5% | 3.272 | $6.342 \times 10^{-17}$ |
| 1000 | 10% | 3.952 | $7.659 \times 10^{-17}$ |
| 3000 | 0% | 1.251 | $2.424 \times 10^{-17}$ |
| 3000 | 5% | 3.270 | $6.338 \times 10^{-17}$ |
| 3000 | 10% | 3.952 | $7.658 \times 10^{-17}$ |

# Acknowledgements

A. G. R. and M. A. H. R. acknowledges support from SNI and PROFOCIE (México).

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
