# Peer review of "Limits on the anomalous magnetic and electric dipole moments of the $\tau$-neutrino in $pp$ collisions at the LHC"

_SciPost Physics Proceedings, doi:SciPost Phys. Proc. 1, 044 (2019)_

## Round 1 · Referee Report · Nicolo De Groot (Referee 1) · 2018-11-26

Strengths

1 - This paper is describing an potentially interesting measurement at the LHC

Weaknesses

1 - The paper suggest it is presenting limits, it is not. It is a feasibility study 2- Some of the fit results are presented without proper explanation 3- The English can be improved

Report

  • I am puzzled by the presence of a photon-neutrino coupling in fig 1
  • The fit results (5), (6), (7) are identical for F2 and F3, this seems odd. It is unclear what is fit to get these results, please explain this in the text.
  • I don't see the added value for reporting both 13 and 14 TeV, since they are similar
  • an annotated version of the paper is included to fix typos etc.

Requested changes

1- Change the title to reflect the fact that this are prospects of a measurement, no actual limits 2- Check the photon-neutrino coupling in fig.1 3- Explain clearly what is fit, is it correct that F2 and F3 are completely identical ? Explain this in the text. 4- Only report 14 TeV and 33 TeV 5- Fix English

Attachment

  • validity: good
  • significance: good
  • originality: high
  • clarity: ok
  • formatting: good
  • grammar: reasonable

Author:  MARIA HERNANDEZ  on 2018-12-13  [id 376]

(in reply to Report 1 by Nicolo De Groot on 2018-11-26)
Category:
correction

I have corrected my contribution

Sincerely,
Maria A. Hernandez-Ruiz

Attachment:

SciPost_A_Gutierrez-Rodriguez-NEW.pdf

Anonymous on 2019-01-11  [id 399]

(in reply to MARIA HERNANDEZ on 2018-12-13 [id 376])
Category:
correction

I correct:

  1. Eqn 5 & 6. You expect to measure a total cross section which will contain a term related to , and also an interference term. How do you isolate the and in this case ? Do you understand why these equations are identical for and ?

Answer to question The cross section of the process pp -> vvgamma + X depends on F_2 and F_3, however to estimate bounds on these parameters we consider F_2 (F_3) one at a time. For this reason, Eqs. (5) and (6) only depend on F_2 (F_3).

Anonymous on 2019-01-10  [id 398]

(in reply to MARIA HERNANDEZ on 2018-12-13 [id 376])
Category:
answer to question

Answer to question

  1. figure 1: The Feynman diagrams contain neutrinos coupling to photons which is wrong. I think only the top two diagrams are physical and in those only when the neutrino couples to a Z.

Answer: The magnetic and electric dipole moments of the neutrino (MM) and (EDM) are one of the most sensitive probes of physics beyond the Standard Model (BSM). On this topic, in the original formulation of the Standard Model neutrinos are massless particles with zero MM. However, in the minimally extended Standard Model containing gauge-singlet right-handed neutrinos, the MM induced by radiative corrections is unobservably small. Similarly, a EDM will also point to new physics and will be of relevance in astrophysics and cosmology, as well as terrestrial neutrino experiments.

Theoretically the electromagnetic properties of neutrinos best studied and well understood are the MM and the EDM. Despite that the neutrino is a neutral particle, neutrinos can interact with a photon through loop (radiative) diagrams. However, a convenient way of studying its electromagnetic properties on a model-independent way is through the effective neutrino-photon interaction vertex which is described by four independent form factors. We study the anomalous MM and the EDM of the tau-neutrino, which are defined in terms of the F_2 and F_3 independent form factor. We are following a focusing as that performed in our previous works [J. F. Nieves, Phys. Rev. D26, 3152 (1982)]. The most general expression for the vertex of interaction neutrino anti-neutrino photon is given by [J. F. Nieves, Phys. Rev. D26, 3152 (1982).] Eq. (1). \begin{eqnarray} \Gamma^{\alpha} & = & eF_{1}(q^{2})\gamma^{\alpha}+\frac{ie}{2m_{\nu_\tau}}F_{2}(q^{2})\sigma^{\alpha \mu}q_{\mu}+ \frac{e}{2m_{\nu_\tau}}F_3(q^2)\gamma_5\sigma^{\alpha\mu}q_\mu \nonumber\ && + \, eF_4(q^2)\gamma_5(\gamma^\alpha-\frac{q\llap{/}q^\alpha}{q^2}), \end{eqnarray}

  1. Eqn 5 & 6. You expect to measure a total cross section which will contain a term related to , and also an interference term. How do you isolate the and in this case ? Do you understand why these equations are identical for and ?

Answer: The cross section of the process pp -> vvgamma + X depends on F_1 and F_2, however to estimate bounds on these parameters we consider F_1 (F_2) one at a time. For this reason, Eqs. (5) and (6) only depend on F_1 (F_2).

Attachment:

SciPost_201810_00003.pdf

Anonymous on 2018-12-14  [id 386]

(in reply to MARIA HERNANDEZ on 2018-12-13 [id 376])
Category:
question

Dear author,

thank you for the improved version, I have 2 remaining issues which I would like to see answered.

  1. figure 1: The Feynman diagrams contain neutrinos coupling to photons which is wrong. I think only the top two diagrams are physical and in those only when the neutrino couples to a Z.

  2. Eqn 5 & 6. You expect to measure a total cross section $\sigma$ which will contain a term related to $F_2^2$, $F_3^2$ and also an interference term. How do you isolate the $F_2$ and $F_3$ in this case ? Do you understand why these equations are identical for $F_2$ and $F_3$ ?

---

## Editorial Decision

published